# First Observation of Embryonic Development and Paralarvae of *Amphioctopus kagoshimensis*

**DOI:** 10.3390/ani15223249

**Published:** 2025-11-10

**Authors:** Jinchao Zhu, Juanwen Yu, Siqing Chen, Tianshi Zhang, Qing Chang, Li Bian

**Affiliations:** 1State Key Laboratory of Mariculture Biobreeding and Sustainable Goods, Yellow Sea Fisheries Research Institute, Chinese Academy of Fishery Sciences, Qingdao 266071, Chinayujw@ysfri.ac.cn (J.Y.);; 2Laboratory for Marine Fisheries Science and Food Production Processes, Qingdao Marine Science and Technology Center, Qingdao 266237, China

**Keywords:** embryonic development, merobenthic octopus, *Amphioctopus kagoshimensis*, paralarval development

## Abstract

**Simple Summary:**

Understanding the early development of cephalopods is essential for advancing aquaculture. *Amphioctopus kagoshimensis*, a merobenthic octopus species, has attracted increasing interest for artificial breeding. However, its reproductive and developmental characteristics remain poorly known. In this study, we successfully induced spawning and observed the complete embryonic development of *A. kagoshimensis* under controlled conditions at 22.0–24.5 °C and salinity of 29–32‰. Each female laid approximately 4000–5000 eggs (2.60 ± 0.05 mm in length). The embryos hatched after 29–30 days, passing through 20 distinguishable developmental stages. Newly hatched paralarvae were planktonic, with a mantle length of 1.58 ± 0.03 mm, and displayed phototactic behavior and ink expulsion. Despite high mortality, a small portion of paralarvae survived over 30 days under current rearing conditions. These results provide the first developmental baseline for this species and will facilitate future efforts in larval rearing, broodstock development, and potential aquaculture of *A. kagoshimensis*.

**Abstract:**

To evaluate the aquaculture potential of *Amphioctopus kagoshimensis*, we investigated the reproductive biology, embryonic development, and early paralarval morphology of *Amphioctopus kagoshimensis* under controlled laboratory conditions. Each adult specimen collected from the coastal waters of Fujian Province spawned approximately 4000–5000 eggs (mean ± SD: 4375 ± 478 eggs), with an overall hatching rate of 75% ± 10% (*n* = 2). Embryonic development lasted approximately 30 days at 22.0–24.5 °C and followed a classical 20-stage pattern. Hatchlings measured an average mantle length of 1.4 ± 0.1 mm and exhibited a merobenthic strategy, characterized by planktonic paralarvae with progressive morphological differentiation. The chromatophores appeared progressively on the head, mantle, arms, and funnel, with numbers increasing from 5 to 23 per arm by 30 days post-hatching. Paralarvae demonstrated active swimming, feeding behavior, and arm sucker development during rearing. By day 30, mantle length reached 2.5 mm, with significant growth in arm length and behavioral complexity. Its relatively small adult size (mantle length 8 cm), a moderate egg size (2.6 mm), fecundity and successful artificial incubation and 30-day paralarvae seedling suggested it may be a suitable model species for developmental studies and potential candidate for merobenthic octopod aquaculture in East Asia.

## 1. Introduction

Octopuses have garnered increasing attention in aquaculture due to their fast growth, high nutritional value, and unique behavioral and reproductive strategies [1,2]. In China, aquaculture efforts for octopuses have been well explored for decades, yet only a few species, notably *Octopus sinensis*, *Octopus minor*, and *Amphioctopus fangsiao*, are currently cultivated at a commercial or experimental scale [3,4,5,6,7,8]. A key limitation in expanding octopus aquaculture lies in their diverse reproductive strategies. Octopuses are broadly classified into two reproductive strategies based on the developmental mode: holobenthic species, which produce few large eggs that hatch directly into benthic juveniles, and merobenthic species, which produce relatively small eggs that hatch into planktonic paralarvae [9,10]. Among these, *O. sinensis* is a typical merobenthic species with high fecundity. However, its artificial rearing remains a major challenge due to high mortality of paralarvae [11,12]. Conversely, *O. minor* and *A. fangsiao* are holobenthic species that produce large eggs yielding benthic juveniles, but their low fecundity and limited scalability restrict their potential for industrial farming [4,8,13]. As a result, only a few octopod species are currently exploited in aquaculture, and the identification of new candidate species for sustainable aquaculture of octopus remains in need.

*Amphioctopus kagoshimensis* (Mollusca: Cephalopoda) was first described by Ortmann in 1888 [14]. Subsequent classification based on both morphological and genetic characteristics has provided further insight into its distinctiveness [15,16]. This species predominantly inhabits the subtropical coastal waters of the Northwest Pacific, with its range extending from southern Japan to Taiwan, and occasional reports from Korea [17]. In Fujian province, southeastern China, *A. kagoshimensis* is locally known as the Kagoshima octopus, regarded as a premium seafood delicacy, and is frequently served at wedding ceremonies and other important occasions. Compared with small-sized species such as Amphioctopus fangsiao (4–6 cm ML) and large merobenthic species such as Octopus vulgaris (20–25 cm ML), *A. kagoshimensis* appears to be a moderately sized species with an adult mantle length of approximately 8 cm [18]. However, there is limited knowledge of its artificial cultivation and experimental studies, particularly on such octopus species of moderate size.

Understanding the reproductive strategy, embryonic development, and paralarval morphological traits of *A. kagoshimensis* is critical for evaluating their aquaculture feasibility. Foundational research by Naef and subsequent contributions by Boletzky [19,20,21,22,23], as well as morphological standards provided by Sweeney, have established standardized stages and terminology for embryonic development [24]. These frameworks have been widely applied to describe the developmental patterns of octopods, including *Amphioctopus aegina* [25], *Octopus maya* [26], *Octopus hubbsorum* [27], and *Paroctopus digueti* [28]. However, the extended post-hatching developmental stages of paralarvae remain poorly understood in most species. Only a limited number of studies have described detailed paralarval morphology beyond hatching, particularly regarding chromatophore patterning, which is an essential trait for species identification and evolutionary studies [29,30].

In this study, we successfully conducted artificial breeding of *A. kagoshimensis* in a laboratory setting, collecting specimens from the coastal waters of Fujian Province. For the first time, we documented the species’ reproductive behavior and embryonic development, providing a detailed description of paralarval morphology. This research lays the groundwork for future biological studies, while offering valuable insights for the potential aquaculture of this species.

## 2. Materials and Methods

### 2.1. Sampling and Cultivation of Wild A. kagoshimensis

In July 2024, wild, sexually mature *A. kagoshimensis* (*n* = 12, male/female ratio = 1:1, wet body weight = 217.5 ± 34.2 g, total length =35.9 to 40.2 cm) were collected from the coastal waters of Zhangzhou, Fujian Province, China (24°30′47″ N 117°38′49″ E). The octopuses were then transported to the aquaculture facility at Fujian Zhangzhou Miaorun Aquatic Co., Ltd. for artificial breeding, where they were reared in an indoor, closed recirculating aquaculture system. The captured octopuses were acclimated in aerated pools (5.0 × 4.0 × 0.5 m) and fed daily with Manila clam (*Ruditapes philippinarum*) for five days. The culture conditions were kept relatively stable, with a water temperature of 21.0–22.5 °C, salinity levels of 29–32, pH ranging from 7.8 to 8.4, and a daily water exchange rate of 100–200%. Temperature (±2 °C; Hailea HC-300A chiller, Hailea Industrial Zone, Guangdong, China), salinity (YSI Pro30 conductivity meter, Yellow Springs, OH, USA), dissolved oxygen (YSI ProODO optical sensor, Yellow Springs, OH, USA), and pH (Hanna HI98121, Hanna Instruments, Woonsocket RI, USA) were monitored daily. Seawater was continuously aerated to maintain dissolved oxygen above 5 mg L^−1^. During acclimation, 12 wild adults (6 ♂:6 ♀; sex ratio 1:1) were kept together in a 5.0 × 4.0 × 0.5 m tank before spawning induction. Before spawning, females were relocated to separate tanks, where we used a custom-designed hatching device (a refuge structure composed of three interconnected PVC buckets, 15 cm × 40 cm each) to incubate the eggs under controlled conditions (Appendix A).

### 2.2. Cultivation and Observation of A. kagoshimensis Paralarvae

After 16 days of rearing, natural mating and spawning occurred. The spawning females (*n* = 4), along with their egg strings and associated shelters, were transferred to a cement pool with continuous flowing water for incubation. During this period, small quantities of Manila clam (*Ruditapes philippinarum*) were provided as daily feed. After 29–30 days of incubation, the hatchlings emerged. The newly hatched hatchlings were then transferred to a larger cement tank (6 m × 5 m × 1.0 m) for continued rearing. During the cultivation period, water temperature ranged from 23.5 °C to 25.0 °C. The seawater was continuously aerated to maintain dissolved oxygen levels above 5.0 mg/L, while salinity was kept stable at 30–32‰. The hatchlings were initially kept in still water for the first three days, followed by daily water exchanges, ranging from 30% to 70%, with light intensity maintained below 1000 lx.

From 0 to 20 days, hatchlings were fed *Artemia* nauplii at 3–5 ind mL^−1^. Between 20 and 30 days, they were given a combination of *Artemia* nauplii and live mysid shrimp (3–5 mg wet mass ind^−1^) with a biomass ratio of approximately 1:5 (mysid/*Artemia*), provided twice daily. Daily observations were conducted to monitor the development of the hatchlings. Each day, ten eggs or hatchlings were randomly sampled, and detailed records of their embryonic development and growth stages were documented. These samples were observed using a microscope (Olympus CX23, Olympus Corporation, Tokyo, Japan), and photographs were taken with a Nikon SMZ800 stereoscopic microscope (Nikon Instruments Inc., Tokyo, Japan) to capture the stages of development. The hatchlings were photographed under the stereomicroscope. Prior to imaging, the stereomicroscope was calibrated using both an ocular micrometer and a stage micrometer. During imaging, reference images were captured simultaneously with the ocular micrometer in view. The corresponding scale marks of the stereomicroscope were noted on each image. Post-processing, including conversion, calibration, correction, and cropping, was conducted using the ImageView software (v2.0). All growth analyses were conducted in R (v4.2.2). To avoid part-whole correlations, mantle length (ML) and arm length (AL) were regressed against days post-hatching (dph) using linear models (stats::lm). Total length (TL) was modeled as an exponential function of time to estimate specific growth rate (SGR). A multiple regression (TL ~ ML + AL) was additionally fitted to assess relative contributions while checking collinearity (VIF < 5). Model performance was evaluated using R^2^ and AIC, and 95% confidence intervals were plotted. The full R script is available as Appendix A.

All experiments were conducted following the guidelines outlined by Fiorito and Andrews [31,32].

### 2.3. Sample Preparation for Scanning Electron Microscope (SEM)

Scanning electron microscopy (SEM) observations were conducted on paralarvae at 1, 5, 15, and 30 days post-hatching (dph), with 3–5 individuals examined per stage. Samples were fixed in 2.5% glutaraldehyde for 4 h, followed by rinsing with phosphate-buffered saline (PBS). They were then dehydrated through a graded ethanol series and substituted with isoamyl acetate. Critical point drying was performed using a CO_2_ critical point dryer (Eiko XD-1, Eiko Engineering Co., Ltd., Hitachinaka, Japan). Subsequently, the specimens were sputter-coated with gold using an ion coater (Eiko IB-3, Eiko Engineering Co., Ltd., Hitachinaka, Japan). Observations were conducted using a JEOL JSM-840 scanning electron microscope (JEOL Ltd., Tokyo, Japan).

### 2.4. Declaration of Generative AI in Scientific Writing

We declare that no generative AI tools were used in the creation of this manuscript. AI tools were only used for language improvement. After using this tool, the authors reviewed and edited the content as needed and take full responsibility for the content of the publication.

## 3. Results

### 3.1. Morphology, Reproductive Behavior and Early Survival of A. kagoshimensis

The length of the arms was between 22.7 ± 2.6 cm, and the mantle length and width were 8.0 ± 0.5 cm and 7.0 ± 0.4 cm, respectively, respectively (Figure 1a). The first, second, and third pairs of arms, as well as the dorsal side of the body, were covered with rough, elliptical protrusions. The fourth pair of arms and the ventral side of the body were smooth and flat, with a rich presence of chromatophores.

The reproductive behavior of *A. kagoshimensis* is similar to that of *Octopus sinensis*, encompassing a sequence of activities during the spawning period, including locomotion, feeding, courtship, copulation, spawning, and egg guarding. Adult locomotion was observed in two forms: crawling and swimming, with crawling being predominant. Individuals showed a preference for feeding on *Litopenaeus vannamei* (whiteleg shrimp), *Mytilus edulis* (blue mussel), and *Ruditapes philippinarum* (Manila clam), while also accepting frozen fish. During courtship, males were more active than females, persistently swimming throughout the tank in search of mates, whereas females exhibited limited movement. Receptive females curled their arms around their bodies, enabling males to insert the hectocotylus into the mantle cavity. Instances of two males simultaneously mating with a single female were recorded, with each copulation lasting approximately 1–3 min. After mating, males died approximately 10 days post copulation, while females began spawning approximately 15 days post mating. A total of four females spawned. Prior to spawning, the food intake of female octopuses was significantly reduced due to brooding behavior. During parental care, females ceased feeding entirely. The egg mass structure appeared irregular and complex, with individual egg strands resembling wheat spikes. The length of a single egg strand ranged from 9.0 ± 4.5 cm, containing approximately 70 to 450 eggs (Figure 1b). A total of four spawnings were obtained from six mature females, resulting in a spawning rate of 66.7%. The total number of eggs per individual ranged from 4000 to 5000, with an average of 4375 ± 478 eggs (*n* = 4), totaling an estimated 16,000–20,000 eggs. Under water temperatures of 22.0–24.5 °C, eggs hatched into paralarvae over a period of 30 days with a hatching rate of approximately 75% ± 10% (*n* = 2), and half of the females died during incubation (around 10 days post spawning). Two females died about 7 days after the paralarvae hatched. With approximately 15,000 initial hatchlings, about 10,000 survived to 5 days post-hatching (72.7%), 3400 remained by 15 dph (22.7%), and only 75 individuals survived to 35 dph (0.5%). Complete mortality occurred by 38 dph.

### 3.2. Embryonic Development of A. kagoshimensis

The embryonic development of *A. kagoshimensis* was divided into 20 stages following the description of Naef, which were grouped into six major developmental phases: cleavage, blastula, gastrula, embryogenesis, organogenesis, and hatching (Table 1; Figure 2 and Figure 3) [19]. Notably, two instances of embryonic inversion, where the embryo turns inside out during its development, were observed during this developmental process. Eggs released by the female exhibited a whitish coloration under natural light and displayed an arc-shaped or rod-like morphology. Under microscopic observation, the eggs appeared darker, with an average length of 2.60 ± 0.05 mm and a width of 0.89 ± 0.02 mm (Figure 2a). The yolk, mirroring the shape of the egg, measured approximately 2.53 mm in length and 0.76 mm in width, with an interstitial space of about 0.06 mm between the yolk and the surrounding egg membrane. The egg stalk, which connected the egg to the egg string, was approximately 3.25 ± 0.05 mm in length and 0.05 mm in width (Figure 2a).

#### 3.2.1. Stage I–III (1–3 Days): Cleavage and Gastrulation

The eggs were white, grain-like, with a slightly narrowed egg stalk (Figure 2b). A perivitelline space was evident at both ends of the yolk; the animal pole began to protrude, forming an embryonic disc. During Stage II, cleavage occurred, with discoidal cleavage observed (Figure 2c). The first and third cleavage divisions were meridional, while the second and fourth divisions were equatorial. The embryo entered gastrulation at Stage III, where mesodermal cells formed muscle tissues around the external yolk sac (Figure 2d). The animal pole extended towards the vegetal pole, and the yolk was partially enveloped by the yolk epithelium, which covered approximately one-fifth of the yolk surface.

#### 3.2.2. Stage IV–X (Days 4–11): Start of Organogenesis

From Stage IV onward, the yolk epithelium gradually expanded to cover the yolk mass. By Stage VIII, it had reached approximately five-sixths of the yolk surface (Figure 2e–i). The yolk sac near the animal pole was continuously depleted, and the embryo began to take on an elongated shape. At Stage IX, the first inversion could be observed, causing the yolk sac to adhere tightly to the lower egg membrane, thereby eliminating the surrounding gap. The animal pole shifted towards the egg stalk, and the yolk sac was consumed externally. The developed organ primordium was difficult to identify. During Stage X, yolk sac consumption accelerated, particularly at the egg stalk, resulting in progressive narrowing from bottom to top (Figure 2k). Primordia of structures such as the mantle, eyes, and arms began to develop, indicating the initial phases of organogenesis.

#### 3.2.3. Stage XI–XV (12–20 Days): Organ Development

By Stage XI, a disk-shaped mantle began to form, covering the upper portion of the yolk sac, while the foundations of the eyes and arms became distinguishable. The mantle, eyes, and arms continued to develop (Figure 2l,m). The eight arms adopted a semi-elliptical shape, were similar in size, and were distributed around the yolk sac. Rapid development occurred in the mantle, eyes, and arms (Figure 2(n1–n3)). The primordium of the visceral mass, funnel, statocyst, brain, and mouth emerged. Rhythmic contractions between the yolk and the yolk sac membrane, originating dorsally and disappearing ventrally, were also evident and lasted approximately five seconds. At Stage XIV, the plates of the eyes had turned deep yellow, and early cardiac activity was detected. The dorsal mantle developed more rapidly than the ventral side, covering a larger portion of the visceral mass and associated tissues. The heart began to form, with transparent, colorless fluid visible during its contractions (Figure 3(a1–a3)). At Stage XV, two branchial hearts appeared on each side of the heart, showing asynchronous pulsations. The mantle fully enveloped the visceral mass and surrounding tissues, and the yolk sac was subdivided into internal and external sacs. The lens had developed, appearing colorless and transparent (Figure 3(b1–b3)).

#### 3.2.4. Stage XVI–XIX (Days 21–28): Chromatophores and Inversions

At Stage XVI, embryonic structures became increasingly distinct, with statocysts positioned on both sides of the esophagus. The external yolk sac was rapidly consumed with rhythmic frequency (Figure 3(c1–c3)). At Stage XVII, the late-stage embryo exhibited rapid development (Figure 3(d1–d3)). Pale-yellow chromatophores emerged on the head, dorsal mantle, and ventral surface, particularly concentrated on the head. The arms developed three suckers, decreasing in size from the base to the tip.

At Stage XVIII, the late-stage embryo was capable of movement within the egg, and chromatophores darkened, turning black on the body and head, which expanded into brown patches (Figure 3(e1–e3)). The head had 10–12 spots, the dorsal side displayed 14–17, and the ventral side had 32–38. These chromatophores also responded to light stimulation. Both the body and branchial hearts exhibited strong activity, with gills now appearing branch-like, symmetrically arranged on both sides. Circular closing muscles appeared on the inner side of the mantle, one on each side of the visceral mass. The visceral mass was dark and had developed an ink sac.

At Stage XIX, some embryos underwent a second inversion and accompanied by an abrupt orange coloration in response to stimulation. Chromatophores appeared along the edge of the dorsal mantle, became enlarged and displayed four additional black chromatophores on the funnel (Figure 3(f1–f3)). At these phases, the function of the neuromuscular and defense systems continued to develop.

#### 3.2.5. Stage XX (29–30 Days): Hatching Stage

Hatching occurred at this stage (Figure 3(g1–g6)), with a total length of 3.53 ± 0.02 mm. Upon receiving stimulation, some hatchlings broke through the egg membrane at the egg stalk end. The external yolk sac remained and was fully absorbed within 4–8 h. The paralarvae exhibited phototactic behavior, tilting their head at an approximate 45° toward the light and swimming at the water’s surface. The first pair of arms was ~0.10 mm longer than the other arms. Under natural light, the visceral mass, otoliths, and external yolk sac appeared pure white. Alternating black and yellow chromatophores were visible on the arms, and back edge of the ocular sac appeared dark green under natural light.

### 3.3. Paralarval Development of A. kagoshimensis

The paralarval development of *A. kagoshimensis* progresses through distinct stages characterized by gradual growth and morphological changes (Table 2; Figure 4a–c). To further elucidate which morphological traits contribute most to total length (TL) growth, we analyzed the relationships between total length and both mantle length (ML) and arm length (AL) across developmental stages (Appendix A). TL showed a stronger linear association with ML (r = 0.965) than with AL (r = 0.936), suggesting that mantle growth contributes more to total length increase during early development.

#### 3.3.1. 1-Day Post-Hatching Paralarvae

These paralarvae displayed rudimentary adhesive abilities and engaged in undulating swimming movements. When exposed to light stimuli, they accelerated rapidly up to 20 cm within 1–2 s, often accompanied by chromatic changes and ink expulsion. The ink sac, teardrop-shaped, measured 0.27 × 0.15 mm. Arm suckers ranged from 0.14 to 0.17 mm in diameter (Figure 5(a1–a3)), with concave protrusions arranged in a honeycomb-like pattern under scanning electron microscopy (Figure 6(a1–a3)).

#### 3.3.2. 3-Day Post-Hatching Paralarvae

On the third day after hatching, clear predatory behaviors could be observed as they fed on *Artemia* nauplii. The gills numbered 7–8 pairs, with individual filaments arranged in a spiral morphology. The statocyst measured 0.35 × 0.26 mm, with a statolith of 0.08 × 0.07 mm. Chromatophores on the arms formed alternating black and yellow stripes upon stimulation (Figure 5(b1–b3)).

#### 3.3.3. 5-Day Post-Hatching Paralarvae

At 5 days, the visceral mass transitioned from white to light pink, and the ocular capsule, measuring 0.40 mm, became more visible (Figure 5(c1–c3)). The arms gradually shortened from dorsal to ventral, and the dorsal interocular distance was 0.70 mm, with a ventral distance of 0.67 mm.

#### 3.3.4. 7-Day Post-Hatching Paralarvae

At 7 days, the paralarvae exhibited positive phototaxis, moving toward light sources, and their body color shifted to reddish-brown (Figure 5(d1–d3)). Enhanced adhesive abilities allowed them to cling to surfaces. The ocular lens protruded one-third from the capsule, measuring 0.13 mm in diameter. A semi-transparent beak was observed.

#### 3.3.5. 10-Day Post-Hatching Paralarvae

By 10 days, paralarvae showed agonistic behaviors, including biting became apparent, and locomotion shifted from a vertical stepwise motion to backward movements. The chromatophores contracted into circular shapes on the mantle and elliptical forms on the arms and head (Figure 5(e1–e3)).

#### 3.3.6. 15-Day Post-Hatching Paralarvae

At 15 days, the original four suckers per arm were aligned in a straight row, while the fifth and sixth appeared offset from the alignment (Figure 5(f1–f3) and Figure 6(b1–b3)). The visceral mass showed a metallic sheen, and the gill and heart structures became distinctly visible. The gills formed elliptical loops, with blood flowing through horizontal ducts.

#### 3.3.7. 20-Day Post-Hatching Paralarvae

At 20 days, the body color was bright, and the paralarvae could expel ink 2–4 times consecutively (Figure 5(g1–g3)). The statocyst became elliptical (0.53 × 0.38 mm), with a truncated-cone-shaped statolith (0.11 × 0.10 mm). The gills occupied two-thirds of the mantle plane, and the individual gill filaments were closely spaced.

#### 3.3.8. 30-Day Post-Hatching Paralarvae

At 30 days, the first pair of arms was the longest, followed by progressively shorter arms. Each arm bore eight suckers arranged in an S-shape arrangement (Figure 5(h1–h3)). The ocular lens measured 0.20 mm in diameter, and the fully developed beak supported advanced locomotion. These paralarvae demonstrated improved ink ejection, feeding, and predation skills, indicating a completion of functional development and behavioral competence.

### 3.4. Chromatophores Pattern During Embryonic and Paralarval Development

During the embryonic development of *A. kagoshimensis*, the formation of chromatophores was a gradual and stage-specific process (Table 3). By stage XVII, pale yellow chromatophores appeared on the head, beneath the dorsal mantle, and along the ventral body surface, with the highest density observed on the head. As development progressed to Stage XVIII, chromatophores on the mantle, head, and arms contracted into black spots and expanded into brown. At this stage, there were typically 10–12 chromatophores on the head, 14–17 on the dorsal mantle, 32–38 on the ventral mantle, and 5 on each arm. These chromatophores exhibited expansion and contraction in response to external stimuli such as light, forming distinct pigment patterns.

From stage XIX, some embryos underwent a second reversion. Upon stimulation, the entire body exhibited an orange-red hue. Chromatophores on the arms increased to 7 per arm, displaying a staggered arrangement, while two rows of five chromatophores each developed on the funnel.

By stage XX, the chromatophore system reached peak development. Red and yellow chromatophores appeared on the mantle, head, and arms, with the arms exhibiting the highest density, and the embryos attained the ability to hatch. In 5-day-old paralarvae, the number of chromatophores on each arm increased to 9, with no noticeable increase on the head or mantle. By 15 days post-hatching, arms developed 13 chromatophores, with four newly formed small chromatophores located centrally along the dorsal side of each arm. At 30 days, the number rose to 23 chromatophores per arm (9 large and 14 small), with the smaller ones irregularly distributed. From day 1 to day 30 post-hatching, no significant changes are observed in the number of chromatophores on the head or mantle.

## 4. Discussion

### 4.1. First Successful Artificial Breeding of A. kagoshimensis

This study represented the first comprehensive report of the artificial breeding and paralarval development of *A. kagoshimensis*, spanning from broodstock domestication through embryonic development to paralarvae rearing beyond 30 days. Unlike most studies that terminate at the hatching stage, our research provides a detailed and extended examination of paralarval development, making it a valuable first reference for a new merobenthic octopod species. However, given the limited sample size, these results should be considered preliminary, and further experiments with larger broodstock cohorts are required to validate the reproductive and developmental parameters observed here.

A spawning rate of 66.7% (four spawning out of six) and a hatch rate of 75% were observed, reflecting strong environmental adaptability comparable to economically significant species such as *O. sinensis*, *O. minor*, and *A. fangsiao* [4,33,34]. These characteristics underscore the suitability of *A. kagoshimensis* for artificial aquaculture.

Reproductive behaviors, including one-male-to-one-female and multi-male-to-one-female mating, followed by egg-guarding, fasting, and senescence after hatching, aligned closely with those reported for *O. sinensis* and *Octopus bimaculoides* [12,29,35,36]. The fecundity of *A. kagoshimensis* ranged from 4000 to 5000 eggs per female. While lower than the fecundity of *O. sinensis* (30,000–180,000 eggs) [12] and *O. vulgaris* (100,000–500,000 eggs) [37], it was significantly higher than that of holobenthic species such as *A. fangsiao* (20–60 eggs) [33] and *O. minor* (50–200 eggs; see Table 4) [3,38]. This intermediate fecundity indicates that *A. kagoshimensis* may represent a manageable and promising candidate for hatchery-scale rearing under controlled conditions.

Merobenthic octopuses, like *A. kagoshimensis*, *O. sinensis*, and *O. vulgaris*, share similar planktonic paralarval characteristics, offering opportunities for comparative research. Advances in paralarval rearing techniques for these species can serve as valuable references for others. Moreover, the relatively small adult size of *A. kagoshimensis* compared to *O. sinensis* and *O. vulgaris* significantly reduces rearing costs by requiring less feed, water, and space (Table 4). This makes *A. kagoshimensis* particularly suitable for experimental studies with multiple groups under controlled conditions. Although its fecundity is lower than larger species, it remains sufficient for research and production purposes, further supporting its potential as a model species for merobenthic octopus aquaculture.

### 4.2. Embryonic Development and Comparative Insights with Other Octopus

The embryonic development of *A. kagoshimensis* proceeded through the standard stages of cleavage, blastula, gastrula, embryogenesis, organogenesis, and hatching, consistent with observations in related species like *A. aegina* and *O. vulgaris* [25,37]. Notably, two instances of embryonic inversion were observed in *A. kagoshimensis*. Similar phenomena have been reported in *A. aegina* and *O. vulgaris* [25,57], suggesting that embryonic inversion may optimize yolk redistribution and support developmental success.

Hatching of *A. kagoshimensis* occurred after approximately 30 days at 22.0–24.5 °C, a duration consistent with other merobenthic species such as *Amphioctopus aegina* (18–22 days at 28 °C) [25] and *Octopus vulgaris* (29–49 days at 22–23 °C; see Table 4) [18]. Merobenthic octopuses, characterized by planktonic paralarvae, typically have smaller eggs (<4 mm), which hatch faster due to their reduced yolk reserves, as seen in *A. kagoshimensis* (2.6 mm egg diameter). In contrast, holobenthic species, which hatch as benthic juveniles, generally have larger eggs (>6 mm), enabling extended yolk reserves and longer incubation periods, such as *Octopus bimaculoides* (10–17 mm eggs), which hatches after 55 days at 24 °C and 85 days at 18 °C [54]. Across both groups, hatching time reflects correlations with egg size and incubation temperature. Larger eggs, whether merobenthic or holobenthic, require longer development times, particularly in colder temperatures. For instance, *O. minor* (21–22 mm egg diameter) hatches in 72–89 days at 21–25 °C [38], while *O. dofleini*, adapted to a cold environment, exhibits the longest hatching time (548 days at 5 °C) [24].

Egg size and fecundity show different patterns between merobenthic and holobenthic octopuses, reflecting their different reproductive strategies. In holobenthic species, which produce benthic juveniles, egg size and fecundity generally correlate with adult body size, with larger species producing larger eggs and having higher fecundity. For example, *Octopus briareus* (12 cm ML) produces large eggs (12–13 mm) with a fecundity of 300–700 [48]. However, exceptions exist, such as *O. minor* (8 cm ML), which produces relatively large eggs (21–22 mm) but has a low fecundity (50–200) [38]. In holobenthic species of similar size, an increase in egg size typically reduces fecundity. Merobenthic species, characterized by planktonic paralarvae, display a different pattern, with weaker correlations between adult body size, egg size, and fecundity. For instance, *Octopus vulgaris* (25 cm ML) produces small eggs (1.5–2 mm) with high fecundity (100,000–600,000) [18], demonstrating that larger adult body size is associated with smaller eggs and higher fecundity. However, *Octopus bimaculatus* has larger adult size (20 cm ML) but produces larger eggs (2.5–4 mm) with relatively low fecundity of 20,000 [42]. *A. kagoshimensis* (8 cm adult size), with its moderate egg size (2.6 mm) and fecundity (4000–5000), reflects a balanced reproductive strategy. While its fecundity is lower than some merobenthic species, such as *O. vulgaris*, it surpasses most holobenthic species with similar egg sizes, making *A. kagoshimensis* an excellent candidate for both experimental research and commercial-scale aquaculture. This balance between egg size and fecundity highlights the adaptability and potential for sustainable aquaculture development of this species.

Chromatophore distribution and quantity during embryonic stages were distinct in *A. kagoshimensis*, with 14–17 dorsal chromatophores and 32–38 ventral chromatophores. These values are higher than those observed in species like *O. mimus* (6–11 dorsal, 24–31 ventral) [58], *O. hubbsorum* (7–10 dorsal, 28–34 ventral) [27], and *A.aegine* (10–12 dorsal, 23–32 ventral) [25]. The unique chromatophore patterns in *A. kagoshimensis* could serve as taxonomic markers during early development.

### 4.3. Paralarval Development and Hatchery Applications

The paralarval development of *A. kagoshimensis* exhibited rapid morphological changes, with significant growth in total length (TL), mantle length (ML), and arm length (AL) over the first 30 days post-hatching. Hatchlings measured 3.44 ± 0.08 mm TL and 1.58 mm ML, showing a similar size to that reported for *A. aegina* (1.9 mm ML) [25]. By day 30, *A. kagoshimensis* paralarvae reached 2.03 mm ML and developed eight suckers per arm.

Sucker morphology in *A. kagoshimensis*, characterized by honeycomb-like concave protrusions, may enhance prey capture and adhesion to surfaces. Additionally, alternating black and yellow chromatophore stripes on the arms provide distinguishing features for species identification. These features, combined with the ability of rearing paralarvae beyond the hatching stage, make *A. kagoshimensis* an invaluable model for studying merobenthic octopus paralarval biology.

Despite these advantages, the high mortality rates observed during the planktonic-to-benthic transition remain a significant obstacle, as seen in other species like *O. vulgaris* and *O. sinensis* [5,12,36]. Addressing this challenge requires optimizing diets, tank design, and environmental conditions. The smaller size and lower maintenance costs of *A. kagoshimensis* facilitate cost-effective experimentation, enabling focused research on overcoming these critical challenges.

Recent studies have demonstrated that both diet composition and hydrodynamic environment play crucial roles in paralarval survival. In *O. sinensis*, co-feeding with nutritionally rich crustacean zoeae such as *Portunus trituberculatus* significantly improved survival rates, while the application of an artificial upwelling rearing system further enhanced larval growth and reduced mortality during settlement [59,60]. These findings suggest that for *A. kagoshimensis*, incorporating live prey with higher nutritional value and maintaining gentle upwelling water flow to prevent bottom aggregation could mitigate mortality during the benthic transition. As a newly characterized merobenthic octopus, *A. kagoshimensis* provides a promising model for refining rearing techniques and understanding developmental strategies within the genus *Amphioctopus*. Continued research integrating nutrition, hydrodynamics, and microbial ecology will be essential to establish stable hatchery production and to deepen our understanding of early cephalopod ontogeny.

## 5. Conclusions

This study is the first to report the detailed embryonic development and 30-day paralarvae development of *A. kagoshimensis* and introduced it as a valuable species for both aquaculture and ecological research. Its high adaptability, moderate fecundity, and potential for cost-effective paralarval rearing demonstrated its suitability for commercial and scientific applications. By extending the observation period beyond 30 days, this study provided one of the most comprehensive accounts of paralarval development in merobenthic octopuses, paving the way for future research and technological advancements.

Continued efforts should focus on optimizing rearing protocols, particularly during the critical planktonic-to-benthic transition and exploring the ecological and genetic factors influencing paralarval survival. By addressing these challenges, *A. kagoshimensis* could serve as a model species for advancing our understanding of octopus biology and promoting sustainable aquaculture practices.

## Figures and Tables

**Figure 1 animals-15-03249-f001:**
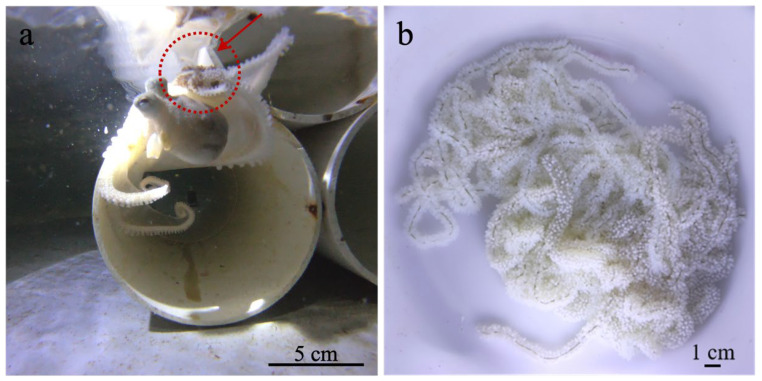
Female *A. kagoshimensis* and its eggs: (**a**) A female *A. kagoshimensis* brooding her eggs (arrow indicating the brooded egg mass). (**b**) Close-up view of the egg string.

**Figure 2 animals-15-03249-f002:**
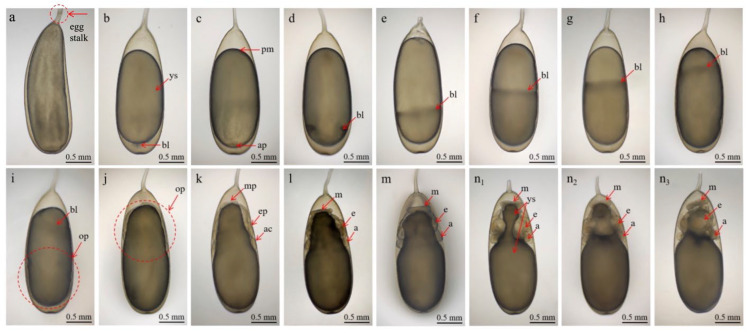
Embryonic development of *A. kagoshimensis*: (**a**) Unlaid egg. (**b**–**n_x_**) Stages I to XIII of embryonic development. (**b**) Stage II (day 2): cleavage; (**c**) Stage III (day 3): gastrulation. “x” representing views: 1 (dorsal view), 2 (ventral view), and 3 (lateral view). Abbreviations: a—arm, ac—arm crown, ap—animal pole, bl—blastoderm, e—eye, ep—eye primordium, m—mantle, mp—mantle primordium, op—organ primordium, pm—perivitelline membrane, ys—yolk sac.

**Figure 3 animals-15-03249-f003:**
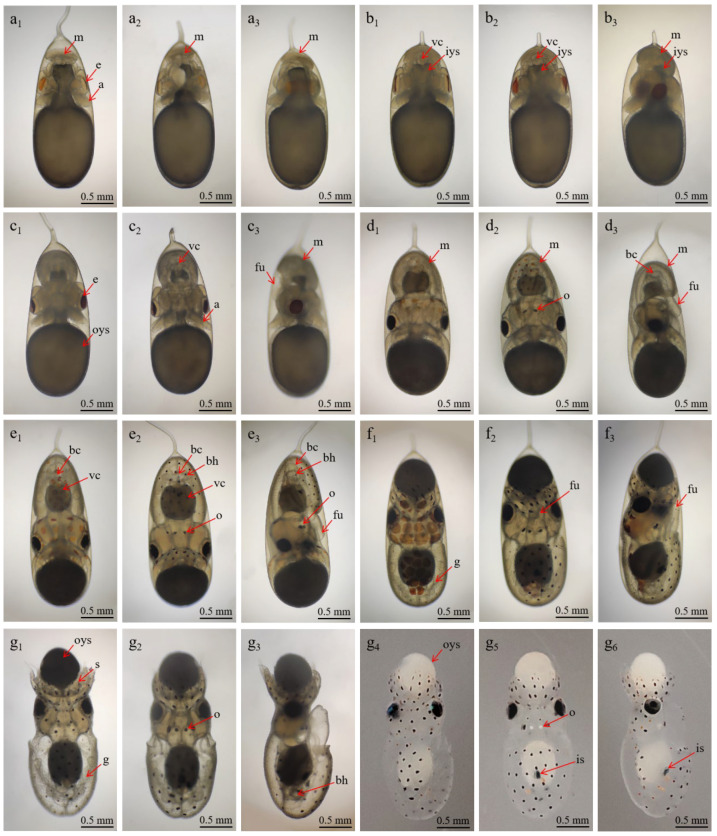
Advanced embryonic development of *A. kagoshimensis*. Panels (**a**–**g**): Depict the stages of embryonic development from Stage XIV to Stage XX. Each stage is represented with up to six views: 1 or 4: dorsal view; 2 or 5: ventral view; 3 or 6: lateral view. Abbreviations: a—arm; bc—body-centered; bh—branchial heart; e—eye; m—mantle; fu—funnel; g—gill; is—ink sac; iys—inner yolk sac; o—otolith; oys—outer yolk sac; s—sucker; vc—visceral mass.

**Figure 4 animals-15-03249-f004:**
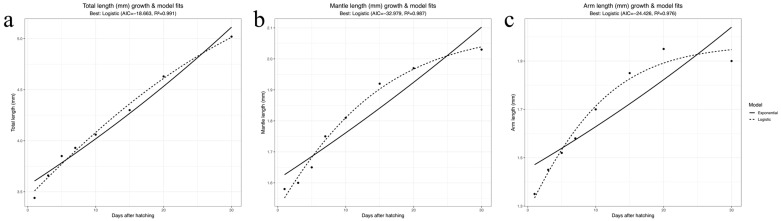
Growth dynamics and morphometric relationships of *A. kagoshimensis* paralarvae. (**a**–**c**) Changes in total length (TL), mantle length (ML), and arm length (AL) during 1–30 days post-hatching (dph), fitted with linear or exponential regression models. Shaded areas indicate 95% confidence intervals. Analyses were performed in R (v4.2.2), and R^2^ values are displayed on each panel. Corresponding TL vs. ML and TL vs. AL plots are provided in Appendix A.

**Figure 5 animals-15-03249-f005:**
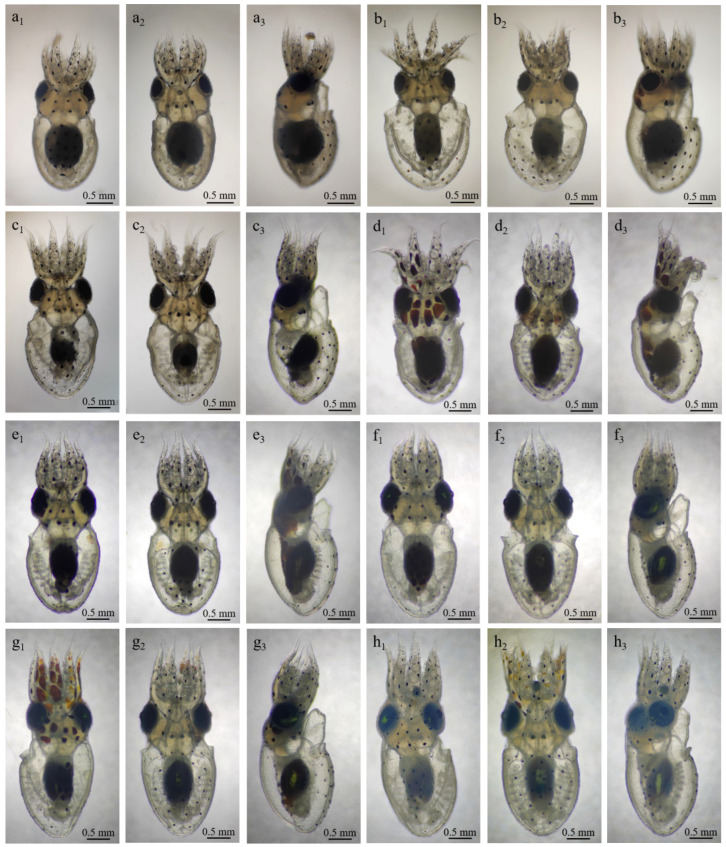
Paralarval development of *A. kagoshimensis*. (**a**–**h_x_**) Stages of paralarval development at 1, 3, 5, 7, 10, 15, 20, and 30 days post-hatching, with “x” representing views: 1 (dorsal view), 2 (ventral view), and 3 (lateral view). Abbreviations: a—arm, bc—body center, bh—branchial heart, e—eye, m—mantle, fu—funnel, g—gill, is—ink sac, iys—inner yolk sac, o—otolith, oys—outer yolk sac, s—sucker, vc—visceral mass.

**Figure 6 animals-15-03249-f006:**
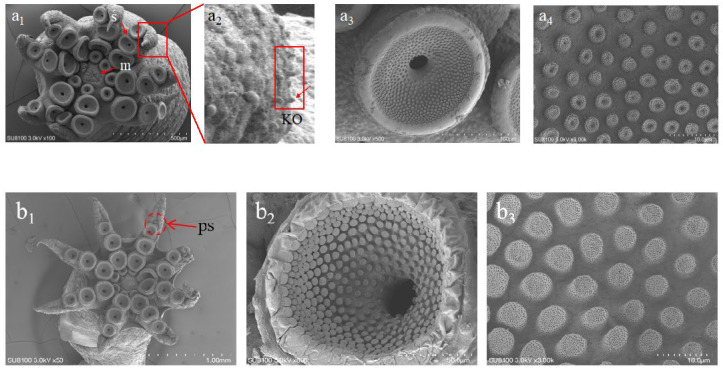
Scanning electron microscopy (SEM) images of arms and suckers in *A. kagoshimensis* paralarvae. (**a_1_**) Arm of a 1-day-old paralarva, showing the arrangement of developing suckers (s) and mouth (m). (**a_2_**) Magnified lateral view of the arm surface from (**a_1_**), highlighting dome-shaped Kölliker organs. (**a_3_**) Sucker of a 1-day-old paralarva. (**a_4_**) Inner surface of the sucker infundibulum of a 1-day-old paralarva. (**b_1_**) Arm of a 15-day-old paralarva, showing well-developed suckers and primordial suckers (ps). (**b_2_**) Sucker of a 15-day-old paralarva. (**b_3_**) Infundibulum of the sucker in a 15-day-old paralarva. Abbreviations: ps—primordial suckers, s—sucker, m—mouth.

**Table 1 animals-15-03249-t001:** Embryonic Development Stages of *A. kagoshimensis*.

Stage	Day (Range)	Key Developmental Features
Stage I	1 day	Eggs appear white and grain-like, with a slightly narrowed egg stalk. Yolk is transparent and elliptical. The eggs measured 2.55 ± 0.10 mm in length and 1.78 ± 0.05 mm in width (Figure 2b).
Stage II	2 days	Cleavage occurs; first and third divisions are meridional, second and fourth are equatorial (Figure 2c).
Stage III	3 days	Gastrulation begins; mesodermal cells form muscle tissues; yolk epithelium partially envelops yolk (Figure 2d).
Stage IV	4 days	Yolk epithelium extends, covering ~1/3 of yolk; narrow longitudinal groove appears (Figure 2e).
Stage V	5 days	Yolk epithelium covers ~1/2 yolk; organogenesis begins but primordia are not yet distinct (Figure 2f).
Stage VI	6 days	Yolk epithelium covers ~2/3 of yolk; yolk becomes darker. Primordia remain indistinct (Figure 2g).
Stage VII	7 days	Yolk epithelium covers ~4/5 of yolk; the yolk sac is largely depleted, causing a visible rut near the animal pole (Figure 2h).
Stage VIII	8 days	Yolk epithelium reaches nearly the vegetal pole, covering 5/6 of yolk; about half of yolk is consumed (Figure 2i).
Stage IX	9 days	First inversion occurs; yolk sac adheres tightly to the egg membrane; animal pole shifts towards the egg stalk (Figure 2j).
Stage X	10–11 days	Yolk sac consumption accelerates; mantle, eyes, and arms begin to develop (Figure 2k).
Stage XI	12–13 days	Yolk sac shortens; disk-shaped mantle begins to form; eyes and arms become distinguishable; mantle disc: 0.44 mm (L) × 0.07 mm (H) (Figure 2l).
Stage XII	14 days	Mantle, eyes, and arms continue to develop; eight arms form symmetrically around yolk sac; mantle height increased ~0.05 mm (Figure 2m).
Stage XIII	15–16 days	Rapid development in mantle, eyes, and arms. Visceral mass, funnel, statocyst, and brain primordia emerge; eye plates ~20 mm diameter; interocular distance ~0.55 mm; arms ~0.23 mm with round tips (Figure 2(n1–n3)).
Stage XIV	17–18 days	Eye plates turn yellow, dorsal mantle develops rapidly. Heart begins to form; eye plates ~0.25 mm; external eye ~0.65 mm; arms ~0.35 mm (Figure 3(a1–a3)).
Stage XV	19–20 days	Two branchial hearts form and function irregularly. The mantle fully envelops the visceral mass. Branchial hearts ~0.11 × 0.09 mm; mantle dorsal length ~0.70 mm, ventral ~0.35 mm; lens ~0.03 mm; arms ~0.50 mm; funnel primordium ~0.35 × 0.15 mm, opening ~0.08 mm (Figure 3(b1–b3)).
Stage XVI	21–22 days	Yolk sac rapidly consumed. Statocyst “D”-shaped ~0.15 × 0.10 mm; arms ~0.60 mm. Mantle length (dorsal/ventral/height): 0.82/0.40/0.55 mm; yolk sac ~1.35 × 1.15 mm (Figure 3(c1–c3)).
Stage XVII	23–24 days	Juvenile reaches 2.84 ± 0.02 mm in total length. Yellow chromatophores appeared on the head, dorsal, and ventral surfaces; body ~1.30 mm; funnel ~0.45 × 0.25 mm; arms with 3 suckers; yolk ~1.10 mm (Figure 3(d1–d3)).
Stage XVIII	25–26 days	Chromatophores form black/brown spots (head: 10–12; dorsal: 14–17; ventral: 32–38); arms ~0.76 mm; circular muscles ~0.22 mm. Strong heart and gill activity (Figure 3(d1–d3)).
Stage XIX	27–28 days	Second inversion occurs; embryos turn bright orange, yolk sac shrinks. Branchial hearts ~0.28 × 0.15 mm; statocysts ~0.30 × 0.28 mm; statoliths ~0.06 mm (Figure 3(e1–e3)).
Stage XX	29–30 days	Paralarvae hatched, phototactic behavior and ink expulsion observed. TL ~3.53 ± 0.02 mm; yolk sac ~0.35 mm (absorbed in 4–8h); black/yellow chromatophores (Figure 3(g1–g6)).

**Table 2 animals-15-03249-t002:** Growth dynamics of *A. kagoshimensis* paralarvae over developmental stages.

Days	1	3	5	7	10	15	20	30
TL (mm)	3.44 ± 0.08	3.66 ± 0.03	3.85 ± 0.02	3.93 ± 0.08	4.06 ± 0.06	4.30 ± 0.07	4.63 ± 0.10	5.02 ± 0.11
ML (mm)	1.58 ± 0.03	1.6 ± 0.02	1.65 ± 003	1.75 ± 0.03	1.81 ± 0.07	1.92 ± 0.10	1.97 ± 0.11	2.03 ± 0.08
MW (mm)	1.4 ± 0.06	1.45 ± 0.04	1.6 ± 0.02	1.63 ± 0.05	1.67 ± 0.07	1.78 ± 0.07	1.82 ± 0.08	1.9 ± 0.10
AL (mm)	1.35 ± 0.04	1.45 ± 0.03	1.52 ± 0.07	1.58 ± 0.05	1.7 ± 0.06	1.85 ± 0.06	1.95 ± 0.08	1.90 ± 0.013
FL (mm)	0.8 ± 0.04	-	-	0.8 ± 0.03	-	-	-	1.15 ± 0.05
FW (mm)	0.65 ± 0.04	-	-	0.65 ± 0.05	-	-	-	0.95 ± 0.08

Abbreviations: TL—Total Length; ML—Mantle Length; MW—Mantle Width; AL—the length of the longest arm; FL—Funnel Length; FW—Funnel Width; -, data not available.

**Table 3 animals-15-03249-t003:** Chromatophores changing during development of *A.kagoshimensis*.

Developmental Stage	Head	Dorsal Mantle	Ventral Mantle	Arms (per Arm)	Funnel
Stage XVIII (embryo)	11 ± 1	15.5 ± 1.5	35 ± 1.5	5	-
Stage XIX (embryo)	11 ± 1	16 ± 1	36 ± 2	7	10 ± 1 (two rows)
Stage XX (hatching)	11 ± 1	16 ± 1	36 ± 2	7	10 ± 1
5 dph paralarvae	11 ± 1	16 ± 1	36 ± 2	9	10 ± 1
15 dph paralarvae	11 ± 1	16 ± 1	36 ± 2	13 ± 1	10 ± 1
30 dph paralarvae	11 ± 1	16 ± 1	36 ± 2	23 ± 2 (9 large + 14 small)	10 ± 1

-, data not available.

**Table 4 animals-15-03249-t004:** Duration of embryonic development and fecundity in octopus species.

Species	Adult Size ML (cm)	Egg Size (mm)	Birth Size ML (mm)	Development Days/ Temperature (°C)	Fecundity (No. Eggs)	Source
*Merobenthic octopus* (Indirect development)						
*Amphioctopus kagoshimensis*	8	2.6	1.58	30/22.0–24.5 °C	4000–5000	Present study
*Octopus rubescens*	8–10	3–4	1.7–2	52/17.7 °C 91/14.8 °C	4000–45,000 1000–19,000	[24,39]
*Octopus insularis*	12	2.13- 2.29	1.68	30–38/26 °C	85,000	[40]
*Octopus vulgaris*	25	1.5–2	2.18 2.4 2–3	29–49/22–23 °C 87/17 °C 15–28/27 °C	100,000–600,000 100,000–500,000	[18,24,41]
*Octopus bimaculatus*	20	2.5–4	2.6	31/16 °C, 50/19 °C	20,000	[42]
*Octopus dofleini*	36	6–8	3–3.5	548/5 °C	30,000–180,000	[24]
*Octopus hubbsorum*	11	1.6	1.22	45/24–26.5 °C	105,000–144,000	[27]
*Octopus defilippi*	9	1.5–2.1	1.3–1.5		10,000+	[24]
*Amphioctopus aegina*	5.83	2.6	1.9	18–22/28 °C	5607–13,640	[25]
*Octopus sinensis*	13.3–17.9	2.3–2.7 2.4 ± 0.2	1.87–2.16	21–24/22.4–23.5 °C 25–35/20.4–23.6 °C	30,000–180,000 90,000–130,000	[5,12]
*Enteroctopus dofleini*	40–60	6–11	6–8	155–223/9–13 °C	20,000–180,000	[43,44]
*Enteroctopus megalocyathus*	>30	7.5–12	7–9.55	131–136/14–18 °C 256–280/8 °C	3000–8600	[30]
*Holobenthic octopus* (Direct development)						
*Amphioctopus fangsiao*		10.1	4.12	30/18–25 °C	294–660	[45,46,47]
*Paroctopus digueti*	4–5	7–10 7–8	5.5 4.5–6	38/27 °C 42/25 °C	50–300 50–150	[28,35]
*Octopus joubini*	4.5 3	6–7 6–10 2.5	5.8 5.5 2.5	35–40/25 °C 35–42/25 °C 32–42/22–24 °C	50–200 25–300 136–2400	[48,49,50]
*Octopus fitchi*	4.5	4–10	4	30–45/25 °C	150–300	[24]
*Hapalochlaena* *maculosa*	5.7	6–7 6–9	4	40–50/20 °C 60/20.8–22.5 °C	150 100–200	[51,52]
*Octopus tehuelchus*	4.95	9–12	6.64	120–150/4–19 °C	80+	[53]
*Octopus minor*	8	21–22	8.5–11.5	72–89/21–25 °C	50–200	[38]
*Octopus bimaculoides*	8.5	10–17	6–7 6.5	55/24 °C, 85/18 °C 46–50/23 °C	250–750	[35,54]
*Octopus briareus*	12	12–13 10–14	7	60–70/25 °C	300–700 150–950	[24,48]
*Octopus maya*	12	11–17	7 4–9	45/25 °C 50–65/wild	300–500 500–5 000	[26,55]
*Octopus californicus*	14 12–14	14–17 17.7–24.8	9.9–12.0	74–77/8–10 °C	50–100 200	[24,56]

This table is based on García-Flores with some modifications, including the addition of new species [28].

## Data Availability

All data supporting the findings of this study are available within the article.

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
