# Peer review of "First Observation of Embryonic Development and Paralarvae of Amphioctopus kagoshimensis"

_animals, 2025, doi:10.3390/ani15223249_

Round 1
Reviewer 1 Report
Comments and Suggestions for Authors
General comments
This manuscript entitled “First observation of embryonic development and paralarvae of Amphioctopus kagoshimensis” provides the first detailed account of artificial breeding, embryonic development, and paralarval morphology of A. kagoshimensis. The study is well organized, clearly written, and offers valuable insights into the early ontogeny of a merobenthic octopus species with aquaculture potential.
Overall, the manuscript is suitable for publication after minor revision, mainly involving small clarifications, consistency in terminology, and figure labeling.
Specific comments
- Line 31 (Abstract): The sentence “Adult specimens … were each spawned 4,000–5,000 eggs” is grammatically awkward. Please rephrase as “Each female spawned approximately 4,000–5,000 eggs.”
- Line 106: Clarify how many females were successfully induced to spawn and how many egg batches were used for developmental observation. This will improve reproducibility.
- Results section, embryonic development: Please confirm whether heartbeats and branchial heart pulsations were observed directly under the stereomicroscope or inferred from previous studies. If directly observed, briefly describe the method used.
- Figure 4: Indicate which software was used to perform the regression analyses, such as R or Origin. Including R² values directly on the plots would make the data clearer.
- Lines 340–360: The data on chromatophore counts are highly valuable. Consider adding mean ± SD or a small table summarizing changes in chromatophore number across developmental stages, for example at embryonic Stage XVIII, and 5, 15, and 30 days post-hatching.
- Figures: Confirm that all micrographs include scale bars and indicate orientation (dorsal or ventral view). Some panels appear cropped and lack visible scale references.
- Table 3: Check the alignment of references and ensure consistent use of spacing and formatting for temperature values (for example, “30 / 22.0–24.5 °C”). Verify that the citation numbers are in the correct sequential order.
- Ensure that all Latin genus and species names are italicized consistently in the text, tables, and figure captions, such as A. fangsiao and O. sinensis.
Author Response
Comments 1: Line 31 (Abstract): The sentence “Adult specimens … were each spawned 4,000–5,000 eggs” is grammatically awkward. Please rephrase as “Each female spawned approximately 4,000–5,000 eggs.”
Response 1: Thank you so much. We already corrected in main body (line 30-31).
Comments 2: Line 106: Clarify how many females were successfully induced to spawn and how many egg batches were used for developmental observation. This will improve reproducibility.
Response 2: Thank you so much for suggestions. We add numbers of spawning females in main body, in total 4 females spawned. Each day, ten eggs or hatchlings were randomly observed.
Comments 3: Results section, embryonic development: Please confirm whether heartbeats and branchial heart pulsations were observed directly under the stereomicroscope or inferred from previous studies. If directly observed, briefly describe the method used.
Response 3: Yes, the heartbeats and branchial heart pulsations were observed directly under stereomicroscope. The detailed information were documented in Methods, line 121-124: ". These samples were observed using a microscope (Olympus CX23), and photographs were taken with a Nikon SMZ800 stereoscopic microscope to capture the stages of development."
Comments 4: Figure 4: Indicate which software was used to perform the regression analyses, such as R or Origin. Including R² values directly on the plots would make the data clearer.
Response 4: Thank you so much for advice. We revised the analysis to avoid part–whole correlations and to clarify the statistical approach and software used. In the revised version, mantle length (ML), arm length (AL), and total length (TL)were each regressed against time (days post-hatching) to evaluate growth trajectories. The corresponding TL vs. ML and TL vs. AL plots were removed from the main text and transferred to the Supplementary Materials for reference. All analyses were performed in R (v4.2.2), and R² values are now displayed on each panel of Figure 4. The statistical methods were also added in the Methods section .
Lines 138–145 (Methods 2.2):
“All growth analyses were conducted in R (v4.2.2). To avoid part–whole correlations, mantle length (ML) and arm length (AL) were regressed against days post-hatching (dph) using linear models (stats::lm). Total length (TL) was modeled as an exponential function of time to estimate specific growth rate (SGR). A multiple regression (TL ~ ML + AL) was additionally fitted to assess relative contributions while checking collinearity (VIF < 5). Model performance was evaluated using R² and AIC, and 95% confidence intervals were plotted. The full R script is available as Supplementary Code S1.”
Comments 5. Lines 340–360: The data on chromatophore counts are highly valuable. Consider adding mean ± SD or a small table summarizing changes in chromatophore number across developmental stages, for example at embryonic Stage XVIII, and 5, 15, and 30 days post-hatching.
Response 5: Thank you so much for advice. A new summary table (Table 3) was inserted (Results 3.4) showing mean ± SD of chromatophore numbers at Stage XVIII and 5, 15, 30 dph.
Comments 6: Figures: Confirm that all micrographs include scale bars and indicate orientation (dorsal or ventral view). Some panels appear cropped and lack visible scale references.
Response 6: We verified scale bars on all micrographs and specified dorsal/ventral orientation in each panel and caption.
Comments 7: Table 3: Check the alignment of references and ensure consistent use of spacing and formatting for temperature values (for example, “30 / 22.0–24.5 °C”). Verify that the citation numbers are in the correct sequential order.
Response 7: We standardized temperature formatting to “30 / 22.0–24.5 °C”, realigned reference columns, and verified that in-text and table citation numbers are sequential and consistent.
Comments 8: Ensure that all Latin genus and species names are italicized consistently in the text, tables, and figure captions, such as A. fangsiao and O. sinensis.
Response 8: We performed a manuscript-wide check to italicize all genus/species names (e.g., A. fangsiao, O. sinensis) in text, tables, and captions.
Reviewer 2 Report
Comments and Suggestions for Authors
Review for the paper “First observation of embryonic development and paralarvae of Amphioctopus kagoshimensis” by Jinchao Zhu, Juanwen Yu, Siqing Chen, Tianshi Zhang, Qing Chang, Li Bian submitted to “Animals”.
The authors of this research paper conducted an analysis on the aquaculture potential of Amphioctopus kagoshimensis, focusing on its reproductive biology, embryonic development, and early morphology under controlled laboratory conditions. They investigated adult specimens sourced from Fujian Province, noting that each female spawned a substantial number of eggs with a notable hatching success rate. The embryonic development process lasted around thirty days, adhering to a well-established developmental pattern. The analysis revealed that the paralarvae exhibited a merobenthic lifestyle, which included distinct morphological changes as they grew. The hatchlings displayed progressive differentiation in both structure and behavior. Paralarval activity, including swimming, feeding, and sucker development, was also observed, further indicating a high viability for cultivation.
The results of this study may have important implications for the future of marine aquaculture, particularly in East Asia, where Amphioctopus kagoshimensis could serve as a suitable model for research into octopod development and cultivation practices.
Some revisions are needed to improve the clarity of the paper.
Introduction.
L 69-71. The authors should report specific evidence for this "high market demand," such as local market price data, catch statistics, or economic reports.
L 73. In my opinion, it would be helpful to clarify what they mean by "moderate size" in this context. Providing size comparisons for small, moderate, and large octopus species would help to position A. kagoshimensis among its relatives.
Materials and Methods.
L 103-104. For the device mentioned, I would suggest the authors explain its design and function in detail. A diagram or photo would be super helpful here. They should also clarify the size of the buckets and how they connected to one another.
L 100-102. The authors should report devices they used to monitor and maintain water quality parameters.
L 106. How many males and females were housed together in the spawning pools? What was the sex ratio in the communal tank?
L 117-118. The feeding regimes need more detail. What the concentration of Artemia nauplii was and the weight of the mysid shrimp. Also, what was the Artemia-to-mysid ratio after day 20?
L 133. The authors should specify the developmental stages of the paralarvae that were prepared for SEM. They should also report the sample size (n) for each stage examined with SEM.
Results.
L 149. It would be beneficial for the authors to report the measurements as mean ± SD, like they did with wet body weight.
L 283-285. The authors should explain the statistical reasoning here, as Mantle Length (ML) and Arm Length (AL) are themselves components of Total Length (TL). Correlating a part with the whole can be misleading. They should consider using a different analysis, such as a regression of ML and AL against time or developmental stage, to more accurately determine their relative growth contributions. Also, the method used here (linear regression) should be desribed in the Materials and methods.
L 302. The authors should report the sample size (n) for each SEM observation. For instance, how many paralarvae were examined at 1 dph to confirm this sucker morphology?
L 340. Regarding the "functional maturity" mentioned for 30-day-old paralarvae, I think the authors should clarify that term since it typically relates to reproductive capability.
Discussion
L 388-390. The authors makes a strong conclusion based on a very small sample size (n=6 for spawning, n=2 for hatching). They should acknowledge the preliminary nature of this data and recommend further trials with larger broodstock cohorts to confirm these rates.
L 398-399. The authors should clarify what aspects of "scalable hatchery production" are enabled by a fecundity of 4,000-5,000 eggs, which is on the lower end for merobenthic species.
L 460-462. The authors stated that "By day 30, A. kagoshimensis paralarvae reached 2.03 mm ML and developed eight suckers per arm, exceeding the growth metrics observed in O. vulgaris (4.19 mm ML)". There is a factual error in stating that a 2.03 mm mantle length exceeds a 4.19 mm mantle length; this needs correction.
L 470-471. The authors should discuss hypotheses regarding the causes of this mortality.
Author Response
- Introduction.
Comments 1.1: L 69-71. The authors should report specific evidence for this "high market demand," such as local market price data, catch statistics, or economic reports.
Response 1.1: We appreciate the reviewer’s valuable suggestion. We acknowledge that our previous expression was not sufficiently rigorous. Because no official statistical or scientific reports are currently available for A. kagoshimensismarket data, we have deleted the phrase referring to “high market demand” from the Introduction (Lines 87–90 in the revised manuscript) to ensure accuracy and avoid unsupported statements.
Comments 1.2: L 73. In my opinion, it would be helpful to clarify what they mean by "moderate size" in this context. Providing size comparisons for small, moderate, and large octopus species would help to position A. kagoshimensisamong its relatives.
Response 1.2: Thank you so much for advice. We revised introduction with detailed comparison based on Table 4 (original Table 3). Line 91-95: “Compared with small-sized species such as Amphioctopus fangsiao (4–6 cm ML) and large merobenthic species such as Octopus vulgaris (20–25 cm ML), A. kagoshimensis appears to be a moderately sized species with an adult mantle length of approximately 8 cm.”
- Materials and Methods.
Comments 2.1: L 103-104. For the device mentioned, I would suggest the authors explain its design and function in detail. A diagram or photo would be super helpful here. They should also clarify the size of the buckets and how they connected to one another.
Response 2.1: Thank you so much for your questions. We used a home-made refuge device for spawning, constructed from polyvinyl chloride (PVC) tubes. Although we did not photograph the equipment during the present experiment, a similar refuge used in subsequent Octopus culture trials is shown in the photo provided below (Supplementary Figure S1). The device consists of three PVC cylinders (each 15 cm in diameter and 40 cm in length) tied together with nylon mesh and fixed to the tank bottom, allowing females to enter and attach egg strings inside the tubes. Detailed description has been added to the Methods section (Lines 107–108).
Comments 2.2: L 100-102. The authors should report devices they used to monitor and maintain water quality parameters.
Response 2.2: Thank you so much for advice. We added detailed information of the monitoring devices and parameters in the Method part.
Line 105–108: “Temperature (±2 °C; Hailea HC-300A chiller), salinity (YSI Pro30 conductivity meter), dissolved oxygen (YSI ProODO optical sensor), and pH (Hanna HI98121 meter) were monitored daily. Seawater was continuously aerated to maintain dissolved oxygen above 5 mg L⁻¹.”
Comments 2.3: L 106. How many males and females were housed together in the spawning pools? What was the sex ratio in the communal tank?
Response 2.3: Thank you so much for advice. We clarified the number of males and females maintained together before spawning.
Line 108-109:
“During acclimation, 12 wild adults (6 ♂ : 6 ♀; sex ratio 1 : 1) were kept together in a 5.0 × 4.0 × 0.5 m tank before spawning induction.”
Comments 2.4: L 117-118. The feeding regimes need more detail. What the concentration of Artemia nauplii was and the weight of the mysid shrimp. Also, what was the Artemia-to-mysid ratio after day 20?
Response 2.4: Thank you so much for advice. We revised the feeding description with detailed prey density and biomass ratio.
Line 125–127:
“From 0 to 20 days, hatchlings were fed Artemia nauplii at 3–5 ind mL⁻¹. Between 20 and 30 days, they were given a combination of Artemia nauplii and live mysid shrimp (3–5 mg wet mass ind⁻¹) with a biomass ratio of approximately 1 : 5 (mysid : Artemia), provided twice daily.”
Comments 2.5: L 133. The authors should specify the developmental stages of the paralarvae that were prepared for SEM. They should also report the sample size (n) for each stage examined with SEM.
Response 2.5: Thank you so much for advice. We have added detailed information on the developmental stages and sample sizes of paralarvae examined under SEM.
Line 206–212: " Scanning electron microscopy (SEM) observations were conducted on paralarvae at 1, 5, 15, and 30 days post-hatching (dph), with 3–5 individuals examined per stage ."
- Results.
Comments 3.1: L 149. It would be beneficial for the authors to report the measurements as mean ± SD, like they did with wet body weight.
Response 3.1: Thank you so much for advice. We changed all parameters data to mean ± SD in Results part.
Comments 3.2: L 283-285. The authors should explain the statistical reasoning here, as Mantle Length (ML) and Arm Length (AL) are themselves components of Total Length (TL). Correlating a part with the whole can be misleading. They should consider using a different analysis, such as a regression of ML and AL against time or developmental stage, to more accurately determine their relative growth contributions. Also, the method used here (linear regression) should be desribed in the Materials and methods.
Response 3.2: Thank you so much for advice. We revised the analysis to avoid part–whole correlations and to clarify the statistical approach and software used. In the revised version, mantle length (ML), arm length (AL), and total length (TL)were each regressed against time (days post-hatching) to evaluate growth trajectories. The corresponding TL vs. ML and TL vs. AL plots were removed from the main text and transferred to the Supplementary Materials for reference. All analyses were performed in R (v4.2.2), and R² values are now displayed on each panel of Figure 4. The statistical methods were also added in the Methods section.
Lines 138–145 (Methods 2.2):
“All growth analyses were conducted in R (v4.2.2). To avoid part–whole correlations, mantle length (ML) and arm length (AL) were regressed against days post-hatching (dph) using linear models (stats::lm). Total length (TL) was modeled as an exponential function of time to estimate specific growth rate (SGR). A multiple regression (TL ~ ML + AL) was additionally fitted to assess relative contributions while checking collinearity (VIF < 5). Model performance was evaluated using R² and AIC, and 95% confidence intervals were plotted. The full R script is available as Supplementary Code S1.”
Comments 3.3: L 302. The authors should report the sample size (n) for each SEM observation. For instance, how many paralarvae were examined at 1 dph to confirm this sucker morphology?
Response 3.3: Thank you so much for advice. We have added the sample size for each developmental stage observed by SEM.
Line 206–212: " Scanning electron microscopy (SEM) observations were conducted on paralarvae at 1, 5, 15, and 30 days post-hatching (dph), with 3–5 individuals examined per stage ."
Comments 3.4: L 340. Regarding the "functional maturity" mentioned for 30-day-old paralarvae, I think the authors should clarify that term since it typically relates to reproductive capability.
Response 3.4: Thank you so much for advice. We revised the sentence to clarify that the term referred to behavioral and physiological capability rather than reproduction.
Line 472–476:
“These paralarvae demonstrated improved ink ejection, feeding, and predation skills, indicating a completion of functional development and behavioral competence.”
- Discussion
Comments 4.1: L 388-390. The authors makes a strong conclusion based on a very small sample size (n=6 for spawning, n=2 for hatching). They should acknowledge the preliminary nature of this data and recommend further trials with larger broodstock cohorts to confirm these rates.
Response 4.1: Thank you so much for advice. We revised this paragraph to acknowledge the limited number of spawning and hatching samples, and to emphasize that the present findings should be confirmed by further trials.
Line 407–413:
“Unlike most studies that terminate at the hatching stage, our research provides a detailed and extended examination of paralarval development, making it a valuable first reference for a new merobenthic octopod species. However, given the limited sample size, these results should be considered preliminary, and further experiments with larger broodstock cohorts are required to validate the reproductive and developmental parameters observed here.”
Comments 4.2: L 398-399. The authors should clarify what aspects of "scalable hatchery production" are enabled by a fecundity of 4,000-5,000 eggs, which is on the lower end for merobenthic species.
Response 4.2: Thank you so much for advice. We revised the statement to provide a more balanced interpretation of the fecundity data and to clarify its implication for hatchery-scale cultivation.
Line 424–426:
“The fecundity of A. kagoshimensis ranged from 4,000 to 5,000 eggs per female. While lower than that of O. sinensis(30,000–180,000 eggs) [12] and O. vulgaris (100,000–500,000 eggs) [36], it was markedly higher than that of holobenthic species such as A. fangsiao (20–60 eggs) [32] and O. minor (50–200 eggs) [3,37]. This intermediate fecundity indicates that A. kagoshimensis may represent a manageable and promising candidate for hatchery-scale rearing under controlled conditions.”
Comments 4.3: L 460-462. The authors stated that "By day 30, A. kagoshimensis paralarvae reached 2.03 mm ML and developed eight suckers per arm, exceeding the growth metrics observed in O. vulgaris (4.19 mm ML)". There is a factual error in stating that a 2.03 mm mantle length exceeds a 4.19 mm mantle length; this needs correction.
Response 4.3: Thank you so much for advice. We agree that the previous comparison could be misleading due to differences in rearing conditions among species. To avoid potential confusion, we have deleted this sentence and kept only the description of A. kagoshimensis paralarval growth and sucker development.
Comments 4.4: L 470-471. The authors should discuss hypotheses regarding the causes of this mortality.
Response 4.4: Thank you so much for advice. We added a paragraph discussing potential causes of paralarval mortality and directions for future improvement.
Line 500-512:
Recent studies have demonstrated that both diet composition and hydrodynamic environment play crucial roles in paralarval survival. In O. sinensis, co-feeding with nutritionally rich crustacean zoeae such as Portunus trituberculatussignificantly improved survival rates, while the application of an artificial upwelling rearing system further enhanced larval growth and reduced mortality during settlement [36,37]. These findings suggest that for A. kagoshimensis, incorporating live prey with higher nutritional value and maintaining gentle upwelling water flow to prevent bottom aggregation could mitigate mortality during the benthic transition. As a newly characterized merobenthic octopus, A. kagoshimensis provides a promising model for refining rearing techniques and understanding developmental strategies within the genus Amphioctopus. Continued research integrating nutrition, hydrodynamics, and microbial ecology will be essential to establish stable hatchery production and to deepen our understanding of early cephalopod ontogeny.